# Optimization of Methane Feed and N:C Ratio for Biomass and Polyhydroxybutyrate Production by the Alphaproteobacterial Methanotroph *Methylocystis* sp. Rockwell

**Hem K. Sharma [1], Dominic Sauvageau [2,*] and Lisa Y. Stein [1,*]**

[1] Department of Biological Sciences, CW 405 Biological Sciences Building, University of Alberta, Edmonton, AB T6G 2E9, Canada
[2] Department of Chemical and Materials Engineering, University of Alberta, Edmonton, AB T6G 1H9, Canada
* Correspondence: dominic.sauvageau@ualberta.ca (D.S.); lisa.stein@ualberta.ca (L.Y.S.); Tel.: +1-780-492-8092 (D.S.); Tel.: +1-780-492-4782 (L.Y.S.)

**Abstract:** The consumption of methane and the production of biodegradable polymers using alphaproteobacterial methanotrophs offers a promising strategy to mitigate greenhouse gas emissions and reduce non-biodegradable plastic pollution. This study identified an ideal amount of added methane and N:C ratio in 100 mL batch cultures of the alphaproteobacterial methanotroph *Methylocystis* sp. Rockwell growing in 1-L sealed bottles using Response Surface Methodology (RSM) to achieve both high biomass and high polyhydroxybutyrate (PHB) production. RSM analysis showed achievement of optimal biomass at 474.7 ± 10.1 mg/L in nitrate mineral salts (NMS) medium and 480.0 ± 65.5 mg/L biomass in ammonium mineral salts (AMS) medium with 8 mmol of methane and an N:C ratio of 0.022. However, optimal PHB concentration was achieved with 6 mmol methane at N:C ratios of 0.012 in NMS medium (149.7 ± 16.1 mg/L) and 0.022 in AMS medium (200.3 ± 5.1 mg/L). A multi-objective RSM analysis projected maxima in PHB production and %PHB cell content (based on dry weight) when using 4.88 mmol methane and N:C ratio of 0.016 in NMS cultures, and 6.28 mmol methane and the 0.016 N:C ratio in AMS cultures. Cultures grown under these projected conditions produced 173.7 mg PHB/L with 46.8% PHB cell content in NMS and 196.9 mg/L with 53.1% PHB cell content in AMS. Taken together, these analyses predicted the optimal conditions for growth and PHB production in batch cultures of *Methylocystis* sp. Rockwell and confirmed a preference for ammonium as the N-source for PHB production. This information is valuable for media formulation in industrial scale-up of *Methylocystis* sp. Rockwell in PHB production.

**Keywords:** methanotrophic bacteria; *Methylocystis* sp. Rockwell; N:C ratio; polyhydroxybutyrate; response surface methodology; single carbon bioconversion

## 1. Introduction

The continued rise in atmospheric methane concentration and human reliance on petroleum-based plastics are of grave concern as both have negative environmental impacts and lead to escalating health issues. Thus, there is a need for development and adoption of mitigating technologies. Bioconversion of methane into biodegradable plastics using methanotrophic bacteria is an appealing premise and a rapidly growing industry [1]. Alphaproteobacterial methanotrophs, including species of *Methylocystis* and *Methylosinus*, produce the biodegradable polymer polyhydroxy-butyrate (PHB) during nutrient deprivation, most commonly triggered by nitrogen starvation [2–4]. These bacteria oxidize methane, assimilate formaldehyde via the serine pathway and, under nitrogen limitation, gate acetyl-CoA towards PHB biosynthesis to store excess carbon into intracellular granules.

Alphaproteobacterial methanotrophs can use nitrate, ammonium or nitrogen fixation to support growth and PHB production, although each isolate has varying preference for its N-source [5]. Moreover, the efficiency of growth and PHB production by methanotrophic

bacteria varies with different combinations of carbon (methane/methanol) and nitrogen (ammonium/nitrate/N2 fixation) sources [4,6–8]. Because nitrogen limitation is a common trigger for PHB production, optimization for industrial scale-up of this process can be achieved by determining an appropriate N:C ratio that is sufficient to support growth but limiting enough to support PHB accumulation, and thus avoid a two-stage production process [9]. Generally, nitrate is preferred over ammonium as the N-source for growing methanotrophs due to competitive inhibition of methane monooxygenase enzymes and the production of the toxic intermediates, hydroxylamine, nitric oxide, and nitrite, when methanotrophs oxidize ammonia as a co-substrate [5,10,11]. *Methylocystis* sp. Rockwell has exceptional tolerance to high ammonium concentrations, is sensitive to nitrite, and shows better growth rates and biomass production with ammonium rather than nitrate as the N-source [7,10]. Moreover, *Methylocystis* sp. Rockwell can produce PHB during log-phase growth and produces a higher %PHB content (based on cell dry weight) when grown with ammonium than with nitrate [12].

A study using Response Surface Methodology (RSM) to pinpoint the optimal N:C ratio for growth and PHB production in *Methylosinus trichosporium* OB3b showed that the highest biomass was achieved at N:C of 0.025 - 0.028, whereas highest PHB production was achieved at N:C of 0.017 [9]. This difference in N:C is logical as higher N-availability supports cell growth rather than PHB biosynthesis. Using methanotrophs for biotechnology requires a species-specific experimental approach due to natural variations and preferences in their niches and nutrients [5,7] The aim of the present study was to define the optimal nutritional parameters to support both growth and PHB production by *Methylocystis* sp. Rockwell, which is a promising strain for industrial applications. We used a multi-objective optimization approach [13] to find the optimal methane amount per culture volume and N:C ratio that simultaneously supported growth and PHB biosynthesis in *Methylocystis* sp. Rockwell. This goal was achievable as *Methylocystis* sp. Rockwell produces PHB during exponential growth, even in the absence of nitrogen limitation [12]. Using methane as a carbon source and ammonium or nitrate as the nitrogen source, N:C was optimized using one-variable-at-a-time (OVAT) followed by RSM with full factorial design. Our results confirmed that ammonium is the preferred N-source, and the optimal methane amount and N:C ratio supporting both robust growth and PHB production in batch cultures of *Methylocystis* sp. Rockwell were identified.

## 2. Materials and Methods

### 2.1. Media Preparation and Bacterial Cultivation

*Methylocystis* sp. Rockwell is a methanotrophic isolate from an aquifer in Canoga Park, CA, USA [14]. Cultures (100 mL) were grown and maintained in nitrate mineral salts (NMS) or ammonium mineral salts (AMS) medium [15] in 1-L gas-tight bottles. A 10X stock medium containing 99 mM of $KNO_3$ or $NH_4Cl$ and 10X nutrient medium were diluted with deionized water to achieve the desired concentrations of N-source and other essential nutrients for each set of experiments. A total of 1 mL sterilized phosphate buffer (26 g of $KH_2PO_4$, 33 g of $Na_2HPO_4$, and milli-Q water up to 1000 mL; pH 6.8) was added to 100 mL sterilized and cooled culture medium followed by the addition of 2 mL fresh inoculum (2% *v/v*) into 1-L Kimble bottles. The bottles were kept gas-tight with screw top lids lined with butyl rubber septa. Methane was injected through a 0.22-μm Millex-GS syringe filter unit (Millipore, Burlington, MA, USA; Sigma, St. Louis, MO, USA), and atmospheric pressure was maintained by previously removing the same amount of air as the volume of methane injected.

Cultures were incubated at 30 °C with shaking at 150 rpm (G10 Gyrotory shaker, New Brunswick Scientific, Edison Township, NJ, USA) until reaching stationary phase (Supplementary Figure S1a). Bacterial growth was analyzed by harvesting 500 μL of culture into each well of a 48-well plate and measuring optical density at 540 nm (OD; Multiskan Spectrum, Thermo Fisher Scientific, Waltham, MA, USA). Cell dry weights were determined by centrifuging 25 mL of culture at $10,000 \times g$ (Sorvall Evolution RC, SS-34

rotor, Thermo Fisher Scientific) at 4 °C for 20 min. The cell pellets were resuspended in 10 mL ultrapure milli-Q water, transferred to a tared weigh dish, and dried at 60 °C to a constant mass [9].

## 2.2. Composition of Headspace Gases

Methane, carbon dioxide, and oxygen were measured in the gas headspace of each culture by extracting 100 µL gas with a 250-µL gas-tight syringe (SGE Analytical Science, Victoria, Australia) and injecting it into a gas-chromatograph with thermal conductivity detector (GC-TCD) (GC 8A, Shimadzu, Kyoto, Japan). The GC-TCD was fitted with packed columns: a molecular sieve 5A column (80/100 6 ft, 2 mm ID SS, Restek, Bellefonte, PA, USA) to detect oxygen, and a Hayesep Q 8A column (80/100 6 ft, 2 mm ID SS, Restek) to detect methane and carbon dioxide. The sample injection ports and detector temperatures were kept at 120 °C for both columns; the molecular sieve column temperature was 90 °C with TCD current of 90 mA, and the Hayesep Q column temperature was 20 °C with TCD current of 120 mA. Ultrahigh purity helium (Praxair, Danbury, CT, USA) was the carrier gas at 160 kPa. Analyte standard curves were generated using pure methane, oxygen, and carbon dioxide gases (Praxair).

## 2.3. Experimental Design and Statistical Analysis

Optimization of methane amount and N:C ratio was accomplished using a one-variable-at-a-time (OVAT) approach and a response surface methodology (RSM) based on a full factorial design [16,17]. The conventional OVAT approach was used to determine the range of N:C ratios for RSM experiments, whereas the RSM approach was used to determine a model-based multi-objective optimal condition (MOOC). This procedure was conducted independently for NMS and AMS media. For wider coverage of effects from variable interactions, the RSM experiments were conducted using a $4^k$ full factorial experimental design ($4^2$ = 16 runs) consisting of two independent variables: methane amount and N:C ratio. The responses of interest included biomass yield, PHB concentration, and %PHB cell content on a dry weight basis. The data were visualized in 3D surface plots using square root transformation in quadratic models [16,18,19]. Each surface response was analyzed by analysis of variance (ANOVA), where $p < 0.05$ showed a significant contribution to the fit of the model. Further, post hoc Tukey's test was used to determine significant differences between treatment conditions. The experimental data were used to develop mathematical models on ANOVA using Design Expert software (Stat-Ease Inc., Minneapolis, MN, USA) [17]. The model was then used to predict expected optimal conditions.

## 2.4. PHB Quantification

PHB production was quantified via depolymerization/derivatization and gas chromatography, as described previously [9], but with slight modifications. Briefly, a 10-mL culture sample harvested after 6 days of incubation was placed into a 12-mL screw-capped glass tube. The tube was centrifuged at $4000 \times g$ (Sorvall Evolution RC, SA-600 rotor, Thermo Fisher Scientific) at room temperature for 30 min. Cell pellets were resuspended in 2 mL chloroform. Further digestion of the sample was achieved by adding 1 mL methanol and 1 mL acidified methanol solution (1 mL benzoic acid standard solution mixed in 24 mL methanol and 1.5 mL concentrated sulfuric acid). The reaction mixture was mixed and submerged in a boiling water bath for 5 h. The digestion process depolymerizes PHB to 3-hydroxybutyric acid, which is derivatized by methylation into the volatile methyl 3-hydroxybutyrate. The methyl benzoate formed from the methylation of benzoic acid is used as an internal standard. After 5 h of reaction, the sample was allowed to cool, and 1 mL of deionized water was added, after which the sample was vortexed for 20 s. The sample phases were allowed to separate, and the organic phase (bottom layer) was collected into a glass vial capped with a butyl rubber septum cap. Then, 3 µL of the sample was injected into a GC-FID (Hewlett Packard, Palo Alto, CA, USA, HP-5890A, Agilent Technologies, Santa Clara County, CA, USA) fitted with a DB-5ms column (30 m × 250 µm × 0.25 µm;

Agilent Technologies). The split ratio was maintained at 1:10, and helium was used as the carrier gas (flow rate of 1.5 mL/min). The injector and detector temperatures were held at 250 °C and 300 °C, respectively. The temperature was programmed at an initial oven temperature of 80 °C, held for 1 min, raised to 120 °C at a rate of 10 °C/min, then to 270 °C at 30 °C/min, and finally held for 3 min. The area ratio of the methyl 3-hydroxybutyrate to methyl benzoate peaks was converted to PHB concentration using a standard curve of crystalline PHB (Sigma-Aldrich) depolymerized and derivatized using the same method.

## 3. Results

### 3.1. One-Variable-at-a-Time (OVAT) Analysis of Biomass and PHB Production

Our study indicated that *Methylocystis* sp. Rockwell had a faster doubling time in batch cultures when grown in ammonium mineral salts (AMS; 8.5 h) versus nitrate mineral salts (NMS; 9.8 h) media (Supplementary Figure S1). Methane was supplied to 100 mL batch cultures at amounts ranging from 2 to 10 mmol. While methane provided at 6 mmol resulted in the highest biomass in both NMS and AMS media, PHB production was optimal when methane was supplied at 8 mmol for NMS and 4 mmol for AMS media (Table 1). We selected 6 mmol methane to pinpoint the optimal N-concentration for PHB production, as this amount led to the highest biomass for both nitrogen sources. Pearson correlation coefficients between the final optical density (OD) and dry weight measurements were 0.985 for values in Table 1 and 0.995 for values in Table 2, showing a positive correlation. Optimal biomass was then achieved with 8 mM nitrate or ammonium, whereas optimal PHB production occurred with 0.5 mM nitrate or 1 mM ammonium (Table 2). Moreover, maximum PHB production was significantly lower ($p$-value = 0.024) for cultures grown with nitrate compared to those with ammonium.

### 3.2. Analysis of Response Surface Methodology (RSM) for Biomass and PHB Production

To explore the combined influence of methane amount and N-source, we created a full factorial design of experiments for RSM analysis. Growth, biomass, and PHB amounts for cultures initiated with methane ranging from 2 to 8 mmol and N:C ranging from 0.002 to 0.032, with either nitrate or ammonium as N-source, were compared using this method. In both NMS and AMS media, methane at 8 mmol and an N:C ratio between 0.022 to 0.032 led to the highest biomass levels (Figure 1). However, different methane amounts and a lower range of N:C ratio (0.012 to 0.022) supported the highest PHB production levels in both NMS and AMS media (Figure 2). Although the optimal %PHB cell content was similar, the total amount of PHB was significantly lower ($p < 0.05$) for cultures grown in NMS than in AMS media (Figure 2).

**Table 1.** Effect of methane amount (mmol) on growth (optical density, OD, at 540 nm), biomass (dry weight, DW, in mg/L), and PHB production (mg/L culture and PHB% of cell dry weight, DW) in batch cultures of *Methylocystis* sp. Rockwell grown with either nitrate or ammonium (10 mM) as the N-source (*n* = 3). Values in bold are the highest levels achieved for each set of measurements.

| KNO$_3$ (mM) | NH$_4$Cl (mM) | CH$_4$ (mmol) | Final OD$_{540}$ | Dry wt. (mg/L) | PHB (mg/L) | PHB Cell Content (%DW) |
|---|---|---|---|---|---|---|
| 10 | - | 0 | 0.009 ± 0.001 | 5.33 ± 2.31 | 0.00 | 0.00 |
| 10 | - | 2 | 0.266 ± 0.028 | 157.33 ± 8.33 | 0.00 | 0.00 |
| 10 | - | 4 | 0.417 ± 0.051 | 270.67 ± 22.03 | 9.16 ± 3.33 | 3.47 ± 1.54 |
| 10 | - | 6 | **0.596 ± 0.012** | **378.67 ± 43.88** | 41.70 ± 16.51 | 11.20 ± 4.89 |
| 10 | - | 8 | 0.558 ± 0.098 | 340 ± 24 | **56.75 ± 6.81** | **16.81 ± 2.98** |
| 10 | - | 10 | 0.550 ± 0.045 | 334.67 ± 32.33 | 35.28 ± 10.21 | 10.45 ± 2.49 |
| - | 10 | 0 | 0.014 ± 0.001 | 5.33 ± 2.31 | 0.00 | 0.00 |
| - | 10 | 2 | 0.393 ± 0.015 | 232 ± 10.58 | 19.83 ± 1.08 | 8.55 ± 0.16 |
| - | 10 | 4 | 0.596 ± 0.003 | 320 ± 20.78 | **50.69 ± 13.72** | **15.90 ± 4.33** |
| - | 10 | 6 | **0.679 ± 0.004** | **461.33 ± 26.63** | 12.85 ± 1.31 | 2.86 ± 0.41 |
| - | 10 | 8 | 0.626 ± 0.004 | 397.33 ± 8.33 | 12.09 ± 2.34 | 3.04 ± 0.54 |
| - | 10 | 10 | 0.601 ± 0.013 | 325.33 ± 33.31 | 11.95 ± 0.92 | 3.72 ± 0.70 |

**Table 2.** Effect of nitrogen concentration (mM) on growth (optical density, OD, at 540 nm), biomass (dry weight, DW, in mg/L), and PHB production (mg/L culture and PHB% of cell dry weight, DW) in batch cultures of *Methylocystis* sp. Rockwell initially containing 6 mmol methane (*n* = 3). Values in bold are the highest level for each set of measurements.

| $CH_4$ (mmol) | $NH_4Cl$ (mM) | $KNO_3$ (mM) | Final $OD_{540}$ | Dry wt. (mg/L) | PHB (mg/L) | PHB Cell Content (%DW) |
|---|---|---|---|---|---|---|
| 6 | - | 0 | $0.153 \pm 0.016$ | $81.33 \pm 14.05$ | $24.19 \pm 6.60$ | $30 \pm 7.94$ |
| 6 | - | 0.5 | $0.470 \pm 0.013$ | $274 \pm 8.49$ | $\mathbf{96.69 \pm 2.67}$ | $\mathbf{35.32 \pm 2.07}$ |
| 6 | - | 1 | $0.612 \pm 0.025$ | $408 \pm 6.93$ | $65.06 \pm 29.82$ | $16.03 \pm 7.60$ |
| 6 | - | 2 | $0.635 \pm 0.015$ | $416 \pm 33.94$ | $18.33 \pm 2.82$ | $4.39 \pm 0.32$ |
| 6 | - | 4 | $0.685 \pm 0.045$ | $466 \pm 59.40$ | $7.34 \pm 3.24$ | $1.63 \pm 0.90$ |
| 6 | - | 8 | $\mathbf{0.731 \pm 0.012}$ | $\mathbf{469.33 \pm 8.33}$ | $10.83 \pm 1.17$ | $2.31 \pm 0.29$ |
| 6 | 0 | - | $0.153 \pm 0.016$ | $81.33 \pm 14.05$ | $24.19 \pm 6.60$ | $30 \pm 7.94$ |
| 6 | 0.5 | - | $0.424 \pm 0.015$ | $276 \pm 10.58$ | $113.31 \pm 9.78$ | $41.15 \pm 4.63$ |
| 6 | 1 | - | $0.621 \pm 0.050$ | $404 \pm 31.24$ | $\mathbf{196.12 \pm 29.75}$ | $\mathbf{48.54 \pm 6.55}$ |
| 6 | 2 | - | $0.668 \pm 0.015$ | $457.33 \pm 25.40$ | $133.62 \pm 26.24$ | $29.20 \pm 5.24$ |
| 6 | 4 | - | $0.720 \pm 0.031$ | $462.67 \pm 25.72$ | $97.15 \pm 12.67$ | $21 \pm 2.37$ |
| 6 | 8 | - | $\mathbf{0.742 \pm 0.003}$ | $\mathbf{465.33 \pm 9.24}$ | $93.33 \pm 5.83$ | $20.05 \pm 1.20$ |

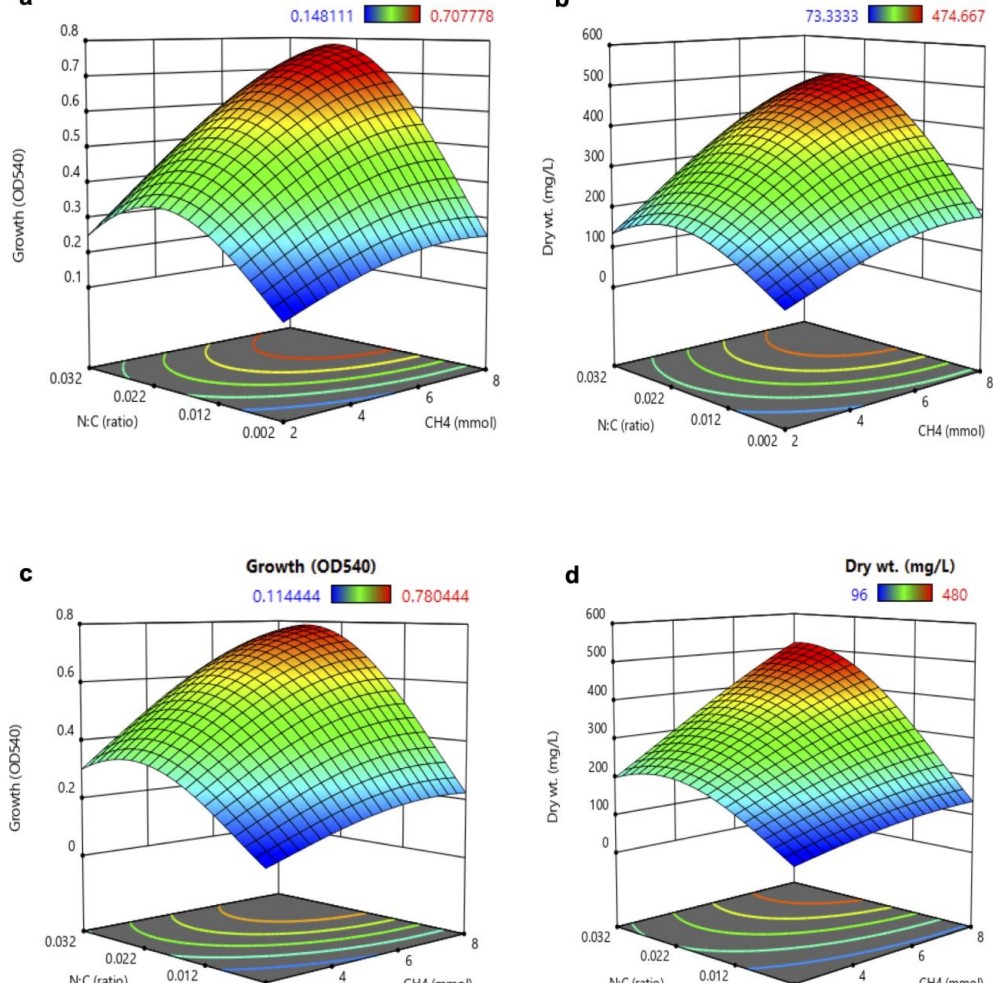

**Figure 1.** RSM results for growth (OD = 540 nm) and biomass (Dry weight in mg/L) in stationary phase batch cultures of *Methylocystis* sp. Rockwell grown with varying amounts of methane against varying N:C ratios. Results are shown for cultures grown in NMS (**a**,**b**) and AMS (**c**,**d**) media (*n* = 3).

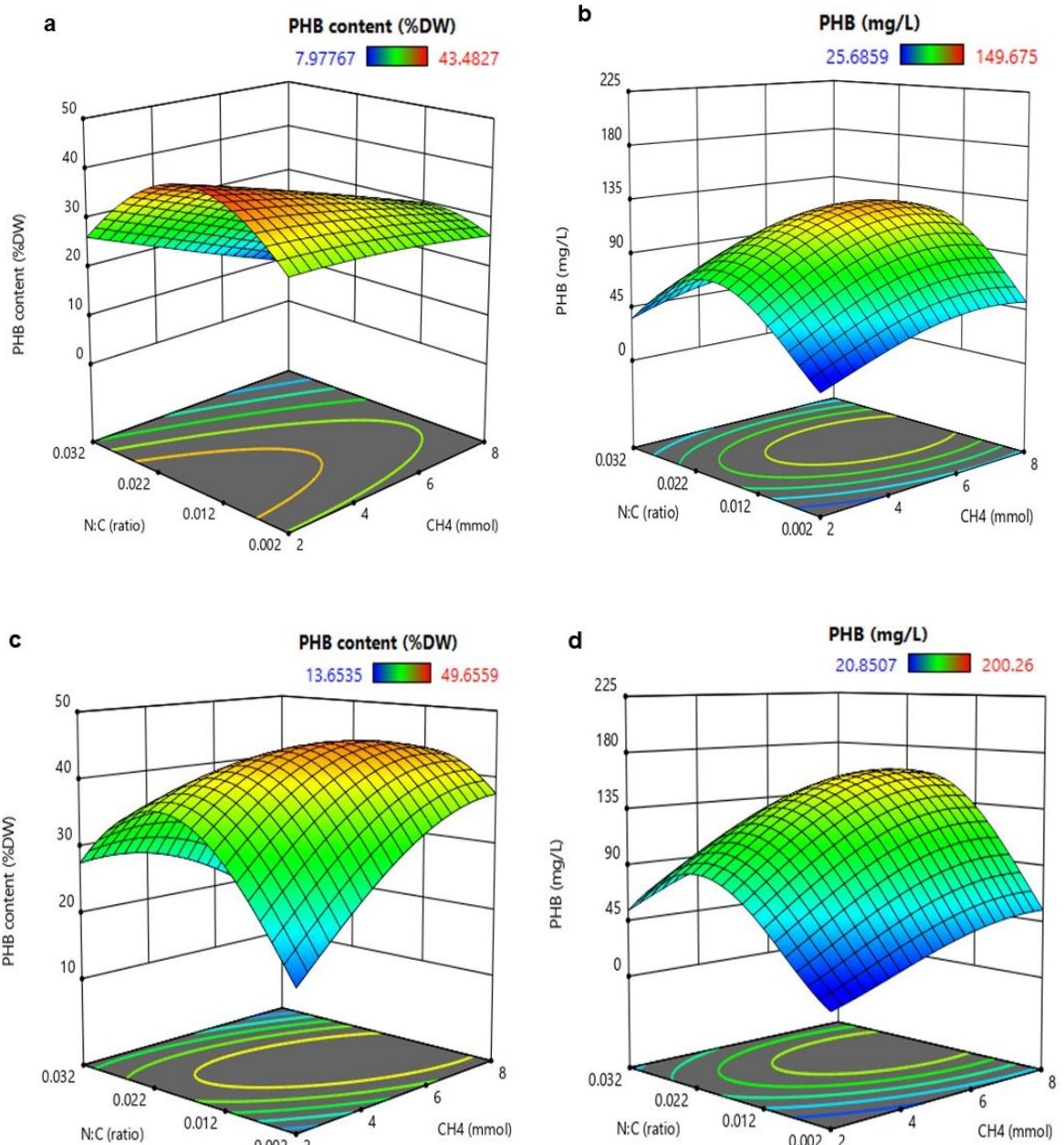

**Figure 2.** RSM results for PHB production (PHB% cell content as % dry weight and PHB amount in mg/L) in stationary phase batch cultures of *Methylocystis* sp. Rockwell grown with varying amounts of methane against varying N:C ratios. Results are shown for cells grown in NMS (**a**,**b**) and AMS (**c**,**d**) media (*n* = 3).

Analysis of the measurements, including final cell dry weight of biomass, PHB concentration, and %PHB cell content, was performed for each medium (AMS or NMS) with one-way ANOVA. The ANOVA of the RSM plots showed that the fit of the quadratic model with square-root transformation was statistically significant ($p < 0.05$). The final model equations for PHB concentration and %PHB content for AMS medium are found in Supplementary Figure S2 and Tables S1–S4, and for NMS medium in Supplementary Figure S3 and Tables S5–S8. Moreover, one-way ANOVA was performed to determine the contributing factors to net PHB concentration using Minitab-16. When *Methylocystis* sp. Rockwell was grown in NMS, the average PHB concentration ranged from 25.7 ± 7.9 to 149.7 ± 16.1 mg/L, and one-way ANOVA showed significant variation among the samples ($p < 0.05$). Further, post hoc Tukey's analyses revealed that PHB concentration of cultures

grown in NMS with 6 mmol of methane and N:C ratio of 0.012 was significantly greater ($p < 0.001$) than all other conditions tested with NMS medium (Supplementary Figure S4a). Similarly, bacteria grown in AMS medium showed an average PHB concentration ranging from $20.9 \pm 2.3$ to $200.3 \pm 5.1$ mg/L, and one-way ANOVA showed a significant difference among the samples ($p < 0.05$). Post hoc Tukey's analyses revealed that PHB concentration for cultures grown in AMS at an N:C ratio of 0.022 and 6 mmol of methane was significantly higher ($p < 0.001$) than all other conditions tested in AMS medium (Supplementary Figure S4b). Together, these results indicate that the optimal methane amount and N:C ratio depended on whether *Methylocystis* sp. Rockwell was grown in NMS or AMS, but overall growth in AMS resulted in both greater biomass and PHB content.

### 3.3. Validation of Multi-Objective Optimal Conditions (MOOC)

Both the OVAT and RSM experiments revealed that, as expected, nitrogen limitation favored PHB production over biomass yield, albeit with differences in PHB yield between nitrate and ammonium as N-source. While PHB concentration was higher at a low N:C ratio (0.012 to 0.022) (Supplementary Figure S4) and bacterial biomass was higher at a high N:C ratio (0.022 to 0.032) (Figure 1), achieving substantial biomass with a high %PHB cell content is crucial for industrial production. Thus, two MOOC analyses were performed with the RSM data to equally balance two parameters: PHB yield and %PHB cell content, or three parameters: biomass yield, PHB yield, and %PHB cell content (Supplementary Figure S5, Table 3). The analysis aimed at achieving optimal PHB yield and %PHB cell content predicted methane at 4.88 and 6.28 mmol for NMS and AMS media, respectively, with an N:C ratio of 0.016. Validation experiments for these conditions showed strong agreement between the predicted and experimental biomass, but significantly higher ($p < 0.05$) experimental PHB concentration than predicted (Table 3). When aiming for high biomass in addition to PHB yield and %PHB cell content, predicted optimal conditions were 6.07 and 6.88 mmol methane for NMS and AMS, respectively, and an N:C ratio of 0.017. However, experimental validation for these conditions showed slightly higher biomass than predicted for AMS, but lower PHB yield for NMS (Table 3). Thus, optimizing for two outcomes (PHB yield, %PHB cell content) led to the best parameterization. The highest biomass ($372 \pm 38.15$ mg/L) with the highest %PHB cell content (53.1% on a cell DW basis) produced by *Methylocystis* sp. Rockwell was similar to values previously found in *M. trichosporium* OB3b (52.5% and 55.5% PHB cell content) [9,20].

**Table 3.** Prediction and experimental validation of multi-objective optimal conditions. Based on the complete set of RSM experimental responses, the multi-objective optimal condition was predicted for NMS and AMS by Design Expert® version-11 software. Optimization #1 was intended to yield maximum %PHB cell content (on the basis of cell dry weight, DW) and PHB concentration, whereas Optimization set #2 was intended to yield maximum biomass, %PHB cell content, and PHB concentration. The projected amount of methane (mmol) and N:C ratios were then tested experimentally for comparison. Standard deviations for $n = 3$ replicates are reported.

| Optimization | N-Source | N:C Ratio | Methane (mmol) | Biomass Yield (mg/L) | | PHB Yield (mg/L) | | PHB Content (% Cell DW) | |
|---|---|---|---|---|---|---|---|---|---|
| | | | | Projection | Experimental | Projection | Experimental | Projection | Experimental |
| 1- %PHB cell content, PHB conc. | NMS | 0.016 | 4.88 | 352.46 | $372 \pm 20$ | 124.44 | $173.65 \pm 13.10$ | 36.08 | $46.79 \pm 4.74$ |
| | AMS | 0.016 | 6.28 | 364.84 | $372 \pm 38.15$ | 158 | $196.93 \pm 12.80$ | 44.50 | $53.11 \pm 3.03$ |
| 2- Biomass, %PHB cell content, PHB conc. | NMS | 0.017 | 6.07 | 414.42 | $431.67 \pm 34.03$ | 130.56 | $102.04 \pm 2.88$ | 32.58 | $23.73 \pm 1.83$ |
| | AMS | 0.019 | 6.88 | 408.66 | $488.33 \pm 7.64$ | 162.88 | $164.44 \pm 3.21$ | 41.72 | $33.68 \pm 0.79$ |

## 4. Discussion

The conversion of methane into PHB varies among the alphaproteobacterial methanotrophs depending on optimal combinations of C- and N-sources [9,21], oxygen availability [9], and trace nutrients like copper [8], among others. It is generally assumed that high N:C leads to high biomass production, while low N:C leads to high PHB accumula-

tion [9]. *Methylocystis* sp. Rockwell differs from most characterized alphaproteobacterial methanotrophs as it prefers ammonium over nitrate as its N-source for growth and PHB production [6,7,10,12]. It also has the ability to produce PHB during the log phase of growth [12] rather than only in the stationary phase (Supplementary Figure S1c).

OVAT and RSM experiments were conducted using a range of methane amounts (added to 100 mL cultures in 1-L bottles) and N:C ratios to evaluate the effects of nutrient sufficiency versus limitation on cell biomass and PHB production with the aim of finding the optimal combination. The initial RSM analysis showed that, with 6 mmol methane, an N:C of 0.022–0.032 was suitable for improving biomass production, whereas an N:C of 0.012–0.022 was more suitable for PHB accumulation. Growth and PHB production were inversely related, as biomass was compromised while PHB production was stimulated under nitrogen limitation, a process regarded as a nutrient stress response [22–24].

A full factorial experimental design combined with RSM analysis projected that a multi-optimization based on PHB concentration and %PHB cell content, but not for biomass, produced the best results for PHB production. While an N:C of 0.016 was found to be optimal for both NMS and AMS media, the optimal amount of methane differed depending on whether nitrate (4.88 mmol methane) or ammonium (6.28 mmol methane) was the N-source. Statistically equivalent ($p < 0.05$) amounts of biomass were produced under these two conditions, but more PHB was produced when ammonium was the N-source.

It should be noted that all experiments in this study were conducted in 1-L bottles to prevent cultures from experiencing oxygen limitation. The high oxygen requirement to achieve optimal biomass and PHB production was similar to that found for *M. trichosporium* OB3b [9]. In smaller bottles (250 mL), 100 mL cultures of *Methylocystis* sp. Rockwell generally reached an $OD_{540}$~0.3 due to oxygen limitation, whereas cultivation of 100 mL culture in 1-L bottles did not experience oxygen limitation and achieved $OD_{540}$~0.7 (Tables 1 and 2). The molar oxygen:methane consumption ratios were consistent at $1.35 \pm 0.05$ for all experiments, which is slightly lower than the oxygen:methane consumption ratio of 1.5 observed for *Methylocystis parvus* cultures [25] and $1.46 \pm 0.05$ for *Methylocystis hirsuta* cultures [26].

Several studies have been conducted to improve PHB production in a variety of alphaproteobacterial methanotrophic strains. In *Methylosinus trichosporium* IMV 3011, improved bacterial growth and PHB production (0.6 g/L) was reported when methanol was added to a methane-grown culture [27]. High PHB concentration (3.43 g/L) was also observed in *M. parvus* OBBP when grown in a continuous-flow reactor [28]. In the present study, under optimal conditions, *Methylocystis* sp. Rockwell at high biomass produced PHB at a similar % cell content (53.1%) as that measured from batch cultures of *M. trichosporium* OB3b (52.5%) under similarly oxygen replete conditions [9]. The projected optimal N:C ratio of 0.016 for *Methylocystis* sp. Rockwell was also in agreement with the optimum N:C ratio of 0.017 observed for *M. trichosporium* OB3b [9], both of which were experimentally confirmed. These results confirm that *Methylocystis* sp. Rockwell can balance growth and PHB production, even in small-scale batch cultures, perhaps due to its ability to produce PHB during exponential growth rather than only in the stationary phase (Supplementary Figure S1c).

## 5. Conclusions

Bioconversion of methane for the production of bioplastics by alphaproteobacterial methanotrophs is an attractive alternative to sugar-based feedstocks as methane is an inexpensive, readily available, non-food-based feedstock, let alone a potent greenhouse gas that must be mitigated. As methanotrophs show distinct preferences for nutrient combinations to support both growth and PHB production, a species-specific MOOC approach can be used to optimize the strain and process toward industrialization. This study highlights *Methylocystis* sp. Rockwell as a promising candidate for industrial PHB production, particularly when using ammonium as an N-source. Fine-tuning of optimal conditions can be further achieved using a continuous gas flow bioreactor to maintain a prolonged exponential growth phase where both biomass and PHB can be produced.

**Supplementary Materials:** The following supporting information can be downloaded at: https://www.mdpi.com/article/10.3390/methane1040026/s1. Figures S1: Growth of Methylocystis sp. Rockwell in nitrate (NMS; red) and ammonium (AMS; blue) mineral salts medium provided with 6 mmol methane gas injected into the headspace of a 100-ml culture in a 1-L bottle; Figures S2: Comparison of the regression model between predicted and actual values for PHB concentration; Figures S3: Comparison of regression models between predicted and actual values for PHB concentration; Figures S4: One-way ANOVA followed by post hoc Tukey's test of PHB production; Figures S5: Model prediction for multi-objectives optimal condition (MOOC) to balance higher PHB yield and %PHB cell content. Tables S1–S4: Effect of methane amount and N:C ratio on PHB production from Methylocystis sp. Rockwell when ammonium is used as the N-source; Tables S5–S8: Effect of methane amount and N:C ratio on PHB production from Methylocystis sp. Rockwell when nitrate is used as the N-source.

**Author Contributions:** L.Y.S. and D.S. envisioned the study, provided funding, supervised the work, and edited the manuscript. L.Y.S., D.S. and H.K.S. designed the experiments. H.K.S. carried out the experiments, analyzed the data, and drafted the manuscript. All authors have read and agreed to the published version of the manuscript.

**Funding:** Funding for this study was provided by the Canada First Research Excellence Fund—Future Energy Systems (CFREF-FES) to D.S. and L.Y.S., the Natural Sciences and Engineering Council of Canada (NSERC)-Discovery Program to D.S. and L.Y.S., the Alberta Innovates BioSolutions-Biofuture Program to D.S. and L.Y.S., and the Alexander Graham Bell Canada Graduate Scholarship-Doctoral (CGSD3-535172) to H.K.S.

**Conflicts of Interest:** The authors declare no conflict of interest.

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
