# Peer review of "Optimization of Methane Feed and N:C Ratio for Biomass and Polyhydroxybutyrate Production by the Alphaproteobacterial Methanotroph Methylocystis sp. Rockwell"

_methane, doi:10.3390/methane1040026_

Round 1

Reviewer 1 Report

The manuscript focuses on the fermentation of Methylocystis sp. using methane as a carbon source to produce PHB. The effects of nitrogen sources were compared, and response surface optimization of different methane amounts and N:C ratios was carried out to obtain the best conditions for PHB production. However, there are the following problems in this manuscript that need to be modified.

1. “The consumption of methane and production of biodegradable polymers by alphaproteobacterial methanotrophs offers a promising strategy to mitigate greenhouse gas emissions and reduce non-biodegradable plastic.” The concentration of methane used in the manuscript was obviously inconsistent with the concentration of methane in the atmosphere. It was suggested to supplement whether the bacteria can effectively metabolize under low methane concentration or atmospheric conditions to ensure that the above problems be solved.

2. This manuscript characterized final OD540 and cell dry weight respectively. However, there was a certain correlation between them. Cell dry weight was also believed to have similar significance with OD value. It is necessary to supplement the difference of these data in the manuscript.

3. It is suggested to keep the maximum ordinate value of b and d in Figure 2 at the same level to enhance the contrast effect.

Author Response

  1. “The consumption of methane and production of biodegradable polymers by alphaproteobacterial methanotrophs offers a promising strategy to mitigate greenhouse gas emissions and reduce non-biodegradable plastic.” The concentration of methane used in the manuscript was obviously inconsistent with the concentration of methane in the atmosphere. It was suggested to supplement whether the bacteria can effectively metabolize under low methane concentration or atmospheric conditions to ensure that the above problems be solved.

We thank the reviewer for this insight. The industrialization of methanotrophs involves their cultivation and maintenance in large-scale (1000's L) bioreactors using methane emissions as a feedstock before its release to the atmosphere, which is not equivalent to using methanotrophs to remove methane from the atmosphere. The idea in the field of methane bioconversion, which has been reviewed numerous times, is to use methane produced from industrial sources, like wastewater treatment plants, instead of flaring or releasing it to the atmosphere. Methane bioconversion and bioplastic production would not be efficient at atmospheric methane concentrations, nor is the industry focused on atmospheric methane as a feedstock. In an industrial context, scale-up of the process using methane as a feedstock can effectively reduce industrial methane emissions and provide a biodegradable source of bioplastics but is not intended to remove atmospheric methane.

  1. This manuscript characterized final OD and cell dry weight respectively. However, there was a certain correlation between them. Cell dry weight was also believed to have similar significance with OD value. It is necessary to supplement the difference of these data in the manuscript.

As the OD value is a rapid, indirect measurement of bacterial growth, the final OD value was used in correlation to the final cell dry weight. We have added the Pearson correlation coefficients between OD and cell dry weight for Tables 1 & 2 to validate this positive correlation (lines 167 – 170).

  1. It is suggested to keep the maximum ordinate value of b and d in Figure 2 at the same level to enhance the contrast effect.

We have incorporated the reviewer’s suggestion in Figure 2.

Reviewer 2 Report

Environmental pollution and the progressive greenhouse effect require measures to prevent these phenomena. All methods should be sought to reduce the burden on the environment by human activities. The manuscript under review contains research results that may be one of the many methods of reducing methane emissions as a greenhouse gas. At the same time, it is possible to obtain biodegradable biomaterials, which is also important from the environmental point of view. However, the question arises: do the authors plan to interest the business with the results of their research? Is it feasible to implement these results on a technical scale?

Author Response

However, the question arises: do the authors plan to interest the business with the results of their research? Is it feasible to implement these results on a technical scale?

We thank the reviewer for this question and have addressed technology for scale-up on line 319-323.

Substantive remarks

Materials and methods:

Chapter 4. Materials and Methods should be moved before Chapter 2. Results. This will allow the manuscript to be organized more logically. Then, the chapters should be renumbered.

We thank the reviewer for this comment and have moved the Materials and Methods section after the Introduction.

Results:

I propose to divide the chapter Results, according to the subtitles marked in bold. The text will become clearer.

We thank the reviewer for this comment and have divided the results section into subsections.

The manuscript does not contain a summary / conclusions. They should be formulated and added after the Discussion chapter.

We thank the reviewer for this comment and have added a brief Conclusion section following the after the Discussion.

Technical notes

Line 15: The abbreviation PHB should be explained.

We have added the definition of PHB on line 15, which is the first use of the PHB abbreviation in the manuscript.

Lines 54-55, 152-153: Remove double hyphens.

We have made this change.

Lines 121-144: In a paragraph, the size and typeface are different and should be standardized.

We have made this change.

Tables 1 and 2: In the right places, please describe the abbreviations DW and OD540

We thank the reviewer for this comment and have defined DW and OD abbreviations in the Tables.

Line 239 The cited Wittenbury et al. 1970 entry is not in References.

We have incorporated this citation.

Line 256 Zaldivar Carrillo should be Zaldivar-Carrillo

We have made this correction.

Supplementary Information

Formula numbers (1-4) should be written on the right side of the text, without dashes.

The changes have been made as recommended.

Reviewer 3 Report

The manuscript lacks novelty

It has nothing new to add to the field of research

The discussion should be enriched 

Where is the predicted model describing the interactive effect of the studied Parameters?

Where is the conclusion?

The abstract and aim did not clarify the novelty of this research paper

Author Response

  1. The manuscript lacks novelty

We have discussed the novelty of our findings on lines 67 – 72.

  1. It has nothing new to add to the field of research

We kindly disagree with this reviewer as there is a large degree of heterogeneity in both carbon and nitrogen preference among methanotrophs that must be defined strain by strain as explained in the Introduction section. As Methylocystis sp. Rockwell has a high potential for industrial applications and scale-up, the information found in this manuscript regarding carbon and nitrogen preference and optimum concentrations is necessary for its further industrialization and will be useful in the development of other methanotrophs as production platforms.

  1. The discussion should be enriched

We believe that the manuscript in its current form is concise and clear. This comment would be helpful if the reviewer could explain deficiencies in the discussion section that could be improved.

  1. Where is the predicted model describing the interactive effect of the studied Parameters?

The model prediction and experimental validations are provided in the main manuscript and supplementary materials. Table 3 clearly provides a comparison between the predicted and experimentally validated results for the models. Furthermore, the interactive effects among the variables are discussed in results section 3.4.

  1. Where is the conclusion?

A new Conclusion paragraph has been added following the discussion.

  1. The abstract and aim did not clarify the novelty of this research paper

We have clarified the novelty of the research on lines 67-72.

Round 2

Reviewer 3 Report

Thanks for doing the required corrections